# Residual Life Prediction of Gas-Engine Turbine Blades Based on Damage Surrogate-Assisted Modeling

**Boris Vasilyev [1,2], Sergei Nikolaev [3,\*] , Mikhail Raevskiy [3] , Sergei Belov [3] and Ighor Uzhinsky [3]**

[1]   Central Institute of Aviation Motors, 111116 Moscow, Russia; bevasilev@ciam.ru
[2]   Faculty of Power Engineering, Department of Gas Turbine Power Plants and Renewable Energy, Bauman Moscow State Technical University, 105005 Moscow, Russia
[3]   Skolkovo Institute of Science and Technology, 121205 Moscow, Russia; Mikhail.Raevskiy@skoltech.ru (M.R.); sergei.belov@skoltech.ru (S.B.); i.uzhinsky@skoltech.ru (I.U.)
\*   Correspondence: s.nikolaev@skoltech.ru

**Abstract:** Blade damage accounts for a substantial part of all failure events occurring at gas-turbine-engine power plants. Current operation and maintenance (O&M) practices typically use preventive maintenance approaches with fixed intervals, which involve high costs for repair and replacement activities, and substantial revenue losses. The recent development and evolution of condition-monitoring techniques and the fact that an increasing number of turbines in operation are equipped with online monitoring systems offer the decision maker a large amount of information on the blades' structural health. So, predictive maintenance becomes feasible. It has the potential to predict the blades' remaining life in order to support O&M decisions for avoiding major failure events. This paper presents a surrogate model and methodology for estimating the remaining life of a turbine blade. The model can be used within a predictive maintenance decision framework to optimize maintenance planning for the blades' lifetime.

**Keywords:** life; remaining useful life; condition-based maintenance; real-time prognostics; surrogate model

---

## 1. Introduction

Gas-turbine engines (GTE) operate in multiregime mode, and their parameters greatly depend on the operating conditions. The actual operating conditions of a GTE may significantly differ from adopted conditions in the design for a typical cycle.

The lifetime of engine components and a host of factors that limit it depend on the scatter of the dimensions, the materials' characteristics, and the working and operating conditions; therefore, it can vary from operator to operator. A life assessment method as a part of condition-based maintenance [1–5] is needed to assess a component's life in order to avoid discarding components that have significant useful life left.

Developing methods for calculating damage to GTE parts in real time is necessary for designing effective prognostic systems. Currently, there are several types of such methods:

1.   Methods based on the use of equations obtained using simplified models; for example, the simple relation between rotor speed and part stress state can be used.
2.   Methods based on big-data analysis and the formation of correlation dependencies between measured parameters and failures [6].

3. Methods based on nondestructive and destructive testing (service-based approach). In this method, it is necessary to find correlation between microstructural degradation and temperature exposure time, and/or service duration [7].

The methods of the first group have several disadvantages:

- The deterioration of engine parameters during operation, and the fact that individual characteristics of both parts and engines are not considered.
- Damage assessment is usually performed for only one critical zone; however, during operation, the critical zone may change.
- Numerical models are usually oversimplified and do not consider the plasticity, creep (stress redistribution), and anisotropy of material characteristics.
- As calculations are performed on the most unfavorable design point, and a substantial safety factor is used to ensure failure-free operation, this causes many components to be discarded too early.

The purpose of this study was to enhance methods of predicting the level of stress and exhausted durability in rotor turbine blades during operation using surrogate-assisted prediction. We propose a machine-learning-based surrogate model that can be efficiently utilized in practice for the estimation of gas-turbine-engine blades' residual life, and thus for their preventive maintenance (Figure 1).

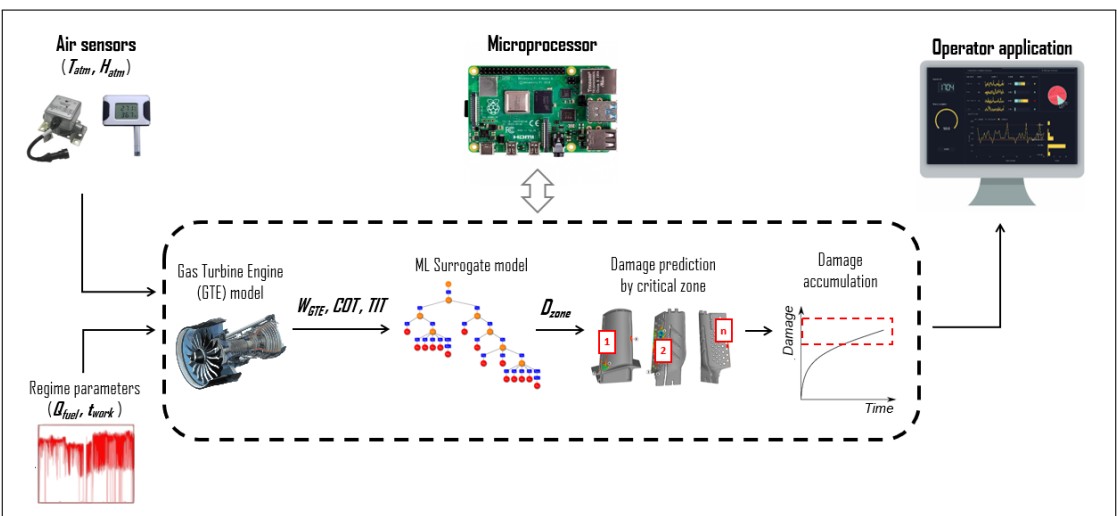

**Figure 1.** Proposed methodology for surrogate model: usage scenario of surrogate model in real operation conditions. $T_{atm}$, atmospheric air temperature; $H_{atm}$, atmospheric air humidity; $Q_{fuel}$, fuel consumption; $t_{work}$, regime duration; $W_{GTE}$, gas-turbine-engine power; $COT$, compressor-outlet temperature; $TIT$, turbine-inlet temperature; $D_{zone}$, separately estimated damage for each critical zone.

## 2. Methods

### 2.1. GTE Model

Figure 2 shows the architecture of the primarily used gas-turbine-engine model. The physics-based model of a gas turbine was separated into submodel blocks that were further sequentially connected. Each submodel is described by equations that define its physics. The applied-physics-based model uses a six-species gas model (oxygen, nitrogen, water vapor, carbon dioxide, argon, and fuel) and requires real operation data to define initial working conditions, such as atmospheric air temperature, pressure and humidity, and pressure in the chambers.

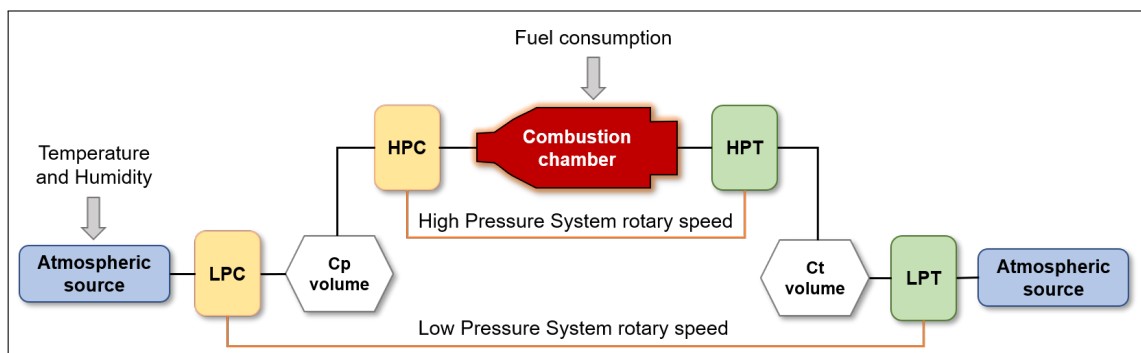

**Figure 2.** Proposed gas-turbine-engine (GTE) model. LPC, low-pressure compressor; HPC, high-pressure compressor; LPT, low-pressure turbine; HPT, high-pressure turbine; Cp volume, chamber with constant volume.

The atmospheric source on its own represents a submodel with zero input and constant thermodynamic conditions. The input parameters that it requires in order to calculate the specific enthalpy and capacity ratio are pressure, temperature, gas-species fraction, and molar mass. Under the first law of thermodynamics for an open system, a submodel of Cp volume (the chamber with constant volume) is then used to solve the variation of internal energy on the basis of the assumption that thermal losses to the environment are negligible [6,8]. Next, the compressor submodel with a performance map that defines several steady-state regimes of the compressor performance was applied to evaluate enthalpy and outlet temperature. The combustion chamber was characterized by the same equations as those of the chamber submodel with a correction for some additional heat produced due to the occurring chemical reaction.

## 2.2. Damage Prediction

Turbine blades operate for extended periods under heavy loads in conditions of nonuniform heating and cyclic loading. Damage is a process that occurs in a material under stress and temperature, and eventually leads to failure [6]. Damage is assumed to be zero ($D = 0$) when the material is new and equal to one ($D = 1$) upon local stress rupture failure. The damage may be due to creep, low-cycle fatigue (LCF), or thermomechanical and high-cycle fatigue, which are some key damage modes. For high-temperature blades, other factors such as oxidation should also be considered [9–11].

By comparing the value of the accumulated damage with its experimentally defined maximal allowable value, one can predict the residual life. Under conditions of the simultaneous action of several damage modes, the linear damage-accumulation (LDA) rule [12,13]) can be used:

$$D_\tau + D_c + D_v + \ldots + D_j = a \tag{1}$$

where $D_\tau$, $D_c$, and $D_v$ characterize the contribution of the effects of static, cyclic, and vibration loads to the total damage, and $a$ is the material parameter. Other damage modes ($D_j$) can also be considered. Although the proposed method allows for us to consider various damage modes, because this study considered a GTE power plant, only static damage was considered:

$$D_\tau = \frac{t_e}{\tau_r} \tag{2}$$

where $t_e$ is the actual time spent under conditions $i$ (local values of temperature and von Mises stress), and $\tau_r$ is the time to failure under condition $i$; $\tau_r$ can be determined using the Larson–Miller parameter (*LMP*) (Equation (3)).

$$\tau_r = 10^{\frac{LMP}{T} - C} \tag{3}$$

where $T$ is absolute temperature (K) and $C$ is the material constant. *LMP* is defined by the value of von Mises stress. Using the LDA rule is less accurate, as the history of the damage is not considered.

During operation in a regime, stress redistribution and relaxation occur (Figure 3). If this process is not considered during strength calculations, the damage value may be several times higher.

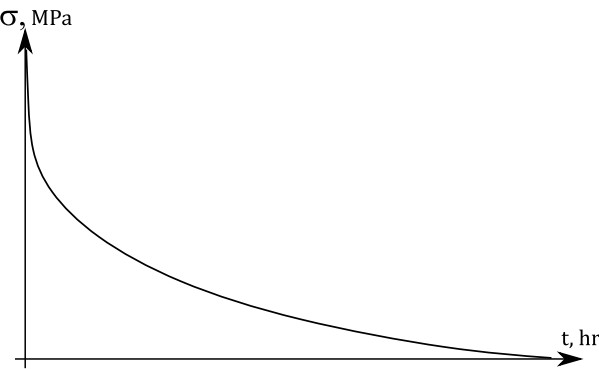

**Figure 3.** Typical relaxation curve.

To calculate the kinetics of blade stress state, strain-hardening theory was used in the ANSYS® software.

$$\dot{\varepsilon}_{cr} = C_1 \cdot \sigma^{C_2} \cdot \varepsilon_{cr}^{C_3} \cdot e^{-C_4/T} \tag{4}$$

where $\dot{\varepsilon}_{cr}$ is creep-strain rate, $\varepsilon_{cr}$ is creep strain, and $C_1...C_4$ are parameters dependent upon blade material.

To calculate damage-taking stress relaxation into consideration, the following equation can be used:

$$D_\tau = \int_0^{t_e} \frac{dt}{\tau_r(t)} \tag{5}$$

where $\tau_r(t)$ is the time to failure at the current temperature ($T$) and stress ($\sigma$) in the blade at moment of time $t$. To calculate the damage, the stress-relaxation curve is divided into a set of segments, at the vertices of which time to failure is determined at the current stress level ($\sigma_{ti}$).

### 2.3. Surrogate-Model Construction

To predict residual life, a machine-learning-assisted surrogate-modeling approach was proposed [1]. This approach is based on a series of 3D stress-state calculations. The obtained surrogate model, along with the GTE model (Figure 1) were used to evaluate the accumulated damage to the turbine blades in real time using the measured parameters. The proposed approach may also be used to determine the influence of operating conditions, such as air inlet parameters, on the engine parts' life. The method consists of the following steps:

1. Selection of parameters and their range that determine the stress state of the blade.
2. Determination of material blade characteristics.
3. Construction of solid-state and finite-element (FE) models of the blade.
4. Calculation of strength with consideration of material anisotropy and nonlinearities.
5. Formation of blade surrogate model.

These steps are described in detail below, some information about the machine learning terms is given in Appendix A.

**Step 1:** The choice of parameters determining the stress state of the blade is based on the GTE model (Figure 1). For high-pressure turbines (*HPT*), such parameters are turbine inlet temperature (*TIT*), RPM, and compressor outlet temperature (*COT*). The range of assigned parameters is based on operation history (including engine prototypes) or on the results of the use of thermodynamic models of the engine.

**Step 2:** Experimental determination of material blade characteristics. The test regimes depend on the type of material; for example, for single-crystal blades, it is necessary to test specimens with various crystallographic orientations [14]. After the experiments, it is necessary to form a set of structural strength characteristics of the material [14]. At this stage, the damage-accumulation rule should also be checked.

**Step 3:** After constructing a solid model of the blade, the FE model is generated (Figure 4) using second-order hexagonal elements consisting of 454,180 nodes and 138,542 elements. ANSYS® software was used to generate mesh and perform calculations (see Appendix B).

After constructing a solid model of the blade, the FE model is generated (Figure 4). It is recommended to use the second-order elements and check the quality of the FE model.

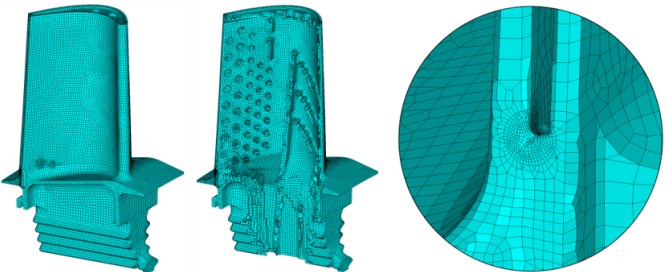

**Figure 4.** Finite-element (FE) model of the considered blade.

**Step 4:** At this stage, the blades' stress–strain state is calculated considering plasticity, creep, and geometric nonlinearity. Depending on the statement of the problem, the contact interaction between blade and disk, and the anisotropy of the material blade characteristics can be considered. For single-crystal turbine blades, the anisotropy of the material characteristics must be considered. For this purpose, a material model is formed with the properties determined by a preliminary study for various crystallographic directions [14]. First, at this stage, preliminary calculations of the blades' strength in several regimes are performed. Using the results of these calculations, the critical locations of the turbine blades are determined (Figure 5).

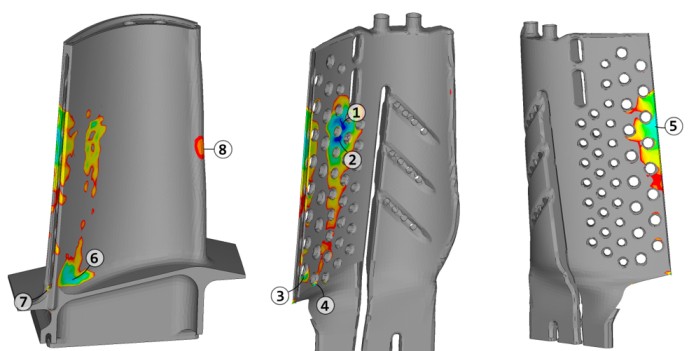

**Figure 5.** Critical zones (1–8) of considered blade.

For *HPT* blades, the complexity of the task (and execution time) can be reduced by using the assumption of maintaining the cooling effectiveness (Θ) when changing the cycle parameters. If the blade's temperature ($T_m$) and the parameters of the cycle on any mode *i* are known, in each of the nodes of the FE model, the cooling effectiveness parameter (Θ) can be calculated as follows:

$$\Theta = \frac{TIT - T_m}{TIT - COT} \tag{6}$$

where *TIT* is the turbine-inlet temperature, $T_m$ is the metal temperature, and *COT* is the compressor-outlet temperature. Calculation of the accumulated damage must take place in real

time; this can be achieved by different methods, including surrogate modeling [15,16]. To form the dependence of the damage on the engine parameters, surrogate modeling can be used, the purpose of which is to build an approximation model to predict the values of the output parameters on the basis of the input parameters from the range of permissible values.

**Step 5:** The surrogate model of the blade is a nonlinear-regression model that is constructed using a combination of ensemble machine-learning methods such as model stacking and boosting. The main steps to construct the proposed surrogate model (Figure 6) were:

1.  Data normalization: Each variable was individually scaled, such that it was in the given range between 0 and 1. The transformation was calculated as follows:

$$X_{norm} = \frac{X - X_{min}}{X_{max} - X_{min}} \tag{7}$$

2.  Regime split and sampling: Since the original target distribution was biased towards near-zero values, which is explained by damage accumulation under usual working conditions, regime split was performed. The data points were split into two groups: working and extreme conditions. Then, the input dataset for the model was obtained by uniform sampling from these two distributions using the bootstrapping technique (random sampling with replacement) to compensate for imbalance between the regimes.
3.  Validation scheme: The resampled dataset from the previous step was divided into training and test sets with partitioning ratios of 80% and 20%, respectively. The training set was then randomly divided into ten subsets in order to perform $k$-fold cross-validation (CV) during model training.
4.  Model training: Each so-called "weak" submodel included in the ensemble was first trained on $(k-1)$ folds of the training data, while the remaining fold was used to make predictions, as well as an evaluation set for early stopping to prevent the submodel from overfitting. The following procedure was repeated $k$ times for each fold. Further, this submodel was fitted on the whole training set, and predictions were then made on the test set. The submodel's predictions from the training set were then used as features to build the master (stacked) model, which in turn was used to make final target predictions on the test set. For each of the models, the root-mean-square error (RMSE) was chosen as an objective function, as well as an accuracy metric during validation.
5.  Model selection: Hyperparameters such as learning rate, maximal tree depth, and number of leaves were optimized for each model using randomized search with independent threefold cross-validation, which showed relatively better performance than that of grid search in finding a global minimum [17].

Code availability: The source code, pretrained ensemble submodels, and full surrogate model were deposited in a GitHub repository (https://github.com/raevskymichail/ciam_ml_model) [18].

The advantages of the proposed approach are:

1.  Several critical zones are tracked.
2.  The life counter can be adapted to a specific part or GTE.
3.  Results of 3D calculations with consideration of plasticity, creep, and material anisotropy can be used.
4.  Loading history can be considered.
5.  It is possible to account for various damage modes and residual stress in parts.
6.  The surrogate model is part of a comprehensive diagnostic solution.

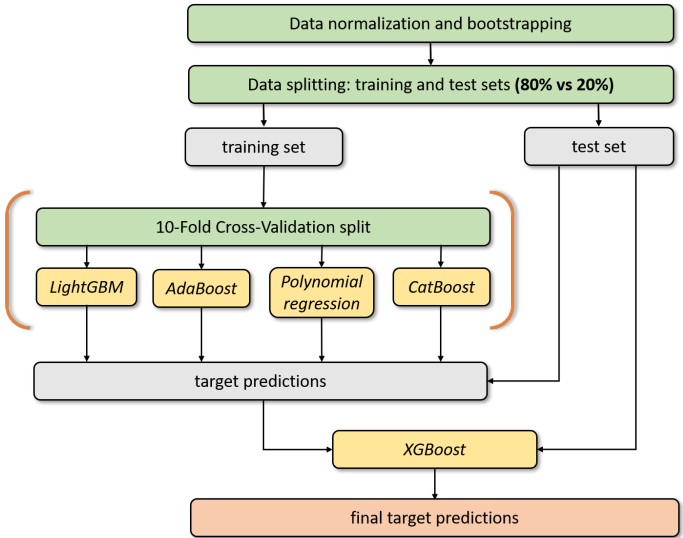

**Figure 6.** Overview of machine-learning-based surrogate model.

## 3. Results and Discussion

Machine-learning-assisted surrogate modeling was successfully applied to a number of engineering problems [19–22]. This encourages its use in optimization and inference methods suited for complex models. Such surrogates can mimic comprehensive physical models while remaining computationally inexpensive, which is often a prerequisite for many engineering applications where available computing resources and maintenance are limited.

In that paper, the HPT rotor blade was considered. Figure 4 shows the FE model of the considered blade. Figure 7 shows the temperature field for one of the regimes.

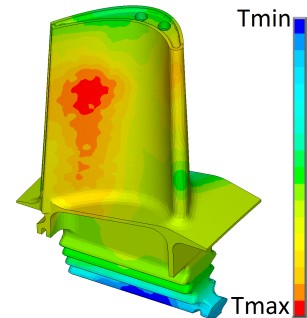

**Figure 7.** Blade-temperature field.

Furthermore, by varying parameters $TIT$, $COT$, and $RPM$ for critical zones using the methods described above, surrogate models for different critical zones were constructed. The constructed surrogate models of the turbine blade, based on boosting and stacking ensemble machine-learning methods, showed satisfactory accuracy, which remained stable for both typical and extreme working conditions (Figure 8).

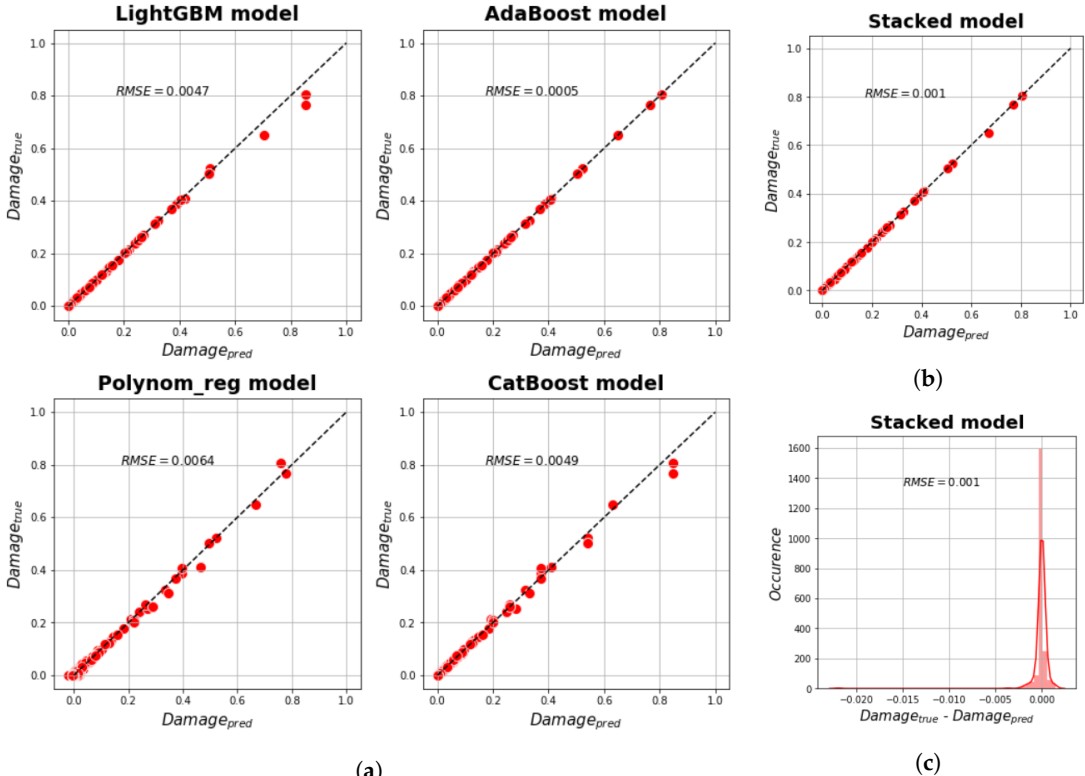

**Figure 8.** Validation results of surrogate model on hold-out test set: (**a**) true vs predicted plot for used submodels; (**b**) true vs predicted plot for master model; (**c**) true minus predicted plot for master model.

To demonstrate the relevance of the approach, we simulated the operation of the considered GTE power plant in two Russian cities. For this purpose, information about the weather over the last five years was used [23]. Figure 9a shows the average temperature values for daily segments. Figure 9b shows a comparison of the accumulation of total damage in one of the critical zones (Zone 6 in Figure 5) of the considered blade. Analysis of the results showed the following:

1. Significant impact of the place of operation. For the considered zone, the total damage value over five years was more than four times higher in the Krasnodar area compared to the Moscow area.
2. A significant increase in damage during the summer period.
3. An increase in the rate of damage accumulation from year to year. For example, the damage rate for 2017 was almost twice as high as that for 2015.

We also considered how linear damage accumulation depends on sampling rate (Figure 10) relative to the damage accumulation obtained on 1 h resolution data using a relative-error (*RE*) formula:

$$RE = \frac{D_i - D_{1h}}{D_{1h}} * 100\% \tag{8}$$

where $D_i$ and $D_{1h}$ are damage accumulation calculated on data with a given sampling rate and on 1 h resolution regime data, respectively.

Result showed that, for the considered GTE data, averaging on a given time step up to 1 month, values had little impact on the resulting linear damage accumulation in comparison with damage accumulation calculated on regime data with a 1 h resolution (*RE* < 0.12% for 5 years of turbine operation).

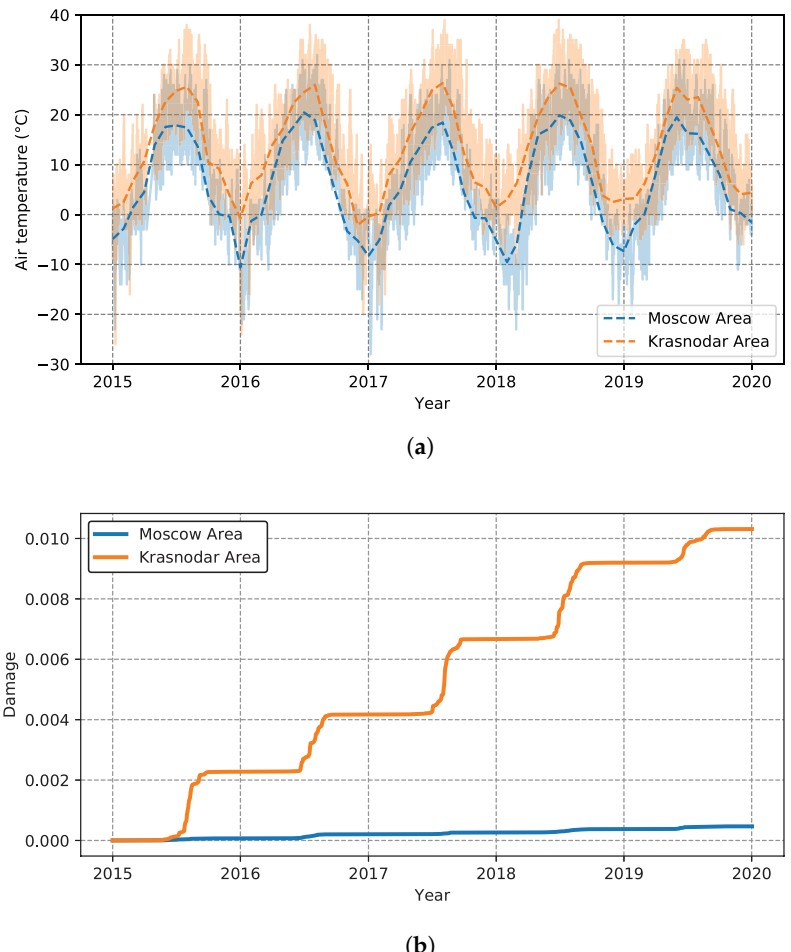

(**a**)

(**b**)

**Figure 9.** Linear damage accumulation from surrogate model for blades working under different temperature conditions: (**a**) average month temperatures in Moscow and Krasnodar areas for 2015–2020 [23]; (**b**) linear damage accumulation in critical Zone 5 (5) over 5 year period of operation.

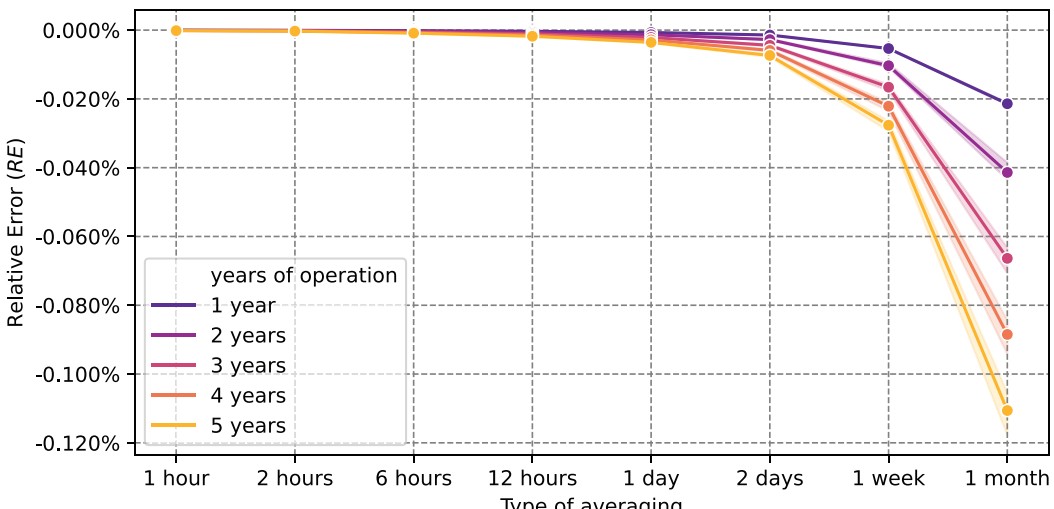

**Figure 10.** Impact of sampling rate on linear damage accumulation. Relative error (RE) between damage accumulation calculated on 1 h data and averaged on a given period.

## 4. Conclusions

In this paper, we proposed a surrogate model based on ensemble machine-learning algorithms that can be utilized in practice for the preventive maintenance of GTE blades via the estimation and monitoring of their residual life. Even though many tasks must be solved for the successful implementation of the proposed algorithm, our method shows the prospects and feasibility of work in this direction. Implementation of the developed approach as part of a comprehensive diagnostic system can help operators make wise maintenance decisions and reduce the likelihood of failure.

## 5. Future Work

The influence of the following phenomena on damage accumulation was not considered:

1.  shutdowns, including emergencies;
2.  transient modes;
3.  scatter in blade dimensions;
4.  the influence of creep and fatigue on each other;
5.  fuel and air quality;
6.  contamination of the turbine gas path.

Each of these phenomena can significantly contribute to the nature of damage accumulation and should be investigated in the future.

Future work is planned:

1.  computational studies of heat-stress state of the unit in transient modes;
2.  increasing the complexity of the used mathematical gas-engine models;
3.  improving the material models; and
4.  Correction factors to consider the acceleration of the damage-accumulation process.

## 6. Grant Information

This work was supported by federal program "Research and development in priority areas for the development of the scientific and technological complex of Russia for 2014–2020" via grant RFMEFI60619X0008.

**Author Contributions:** Methodology, B.V. and S.N.; Software, M.R. and S.B.; Supervision, I.U. All authors have read and agreed to the published version of the manuscript.

**Funding:** This work was supported by the federal program "Research and development in priority areas for the development of the scientific and technological complex of Russia for 2014–2020" via grant RFMEFI60619X0008.

**Acknowledgments:** The authors would like to express their gratitude to colleagues Artem Semenov and Nikita Losyakov for their help with numerical calculations.

**Conflicts of Interest:** Authors declare no conflict of interest.

## Abbreviations

| | |
|---|---|
| *COT* | Compressor outlet temperature |
| FEM | Finite-element model |
| GTE | Gas-turbine engine |
| *HPT* | High-pressure turbine |
| LCF | Low-cycle fatigue |
| LDA | Linear damage accumulation |
| *LMP* | Larson–Miller parameter |
| *RPM* | Revolutions per minute |

## Appendix A. Terminology

Hyperparameter: In machine learning, a parameter of which the value is used to control the learning process. By contrast, the values of other parameters (typically node weights) are derived via training.
K-fold cross-validation: A method for evaluating the analytical model and its behavior on independent data. Available data are divided into k parts. Then, the model is trained on k − 1 parts of the data, and the rest of the data are used for testing. The procedure is repeated k times; in the end, each k piece of data is used for testing. The result is an assessment of the effectiveness of the selected model with the most uniform use of available data.
Ensemble learning: In statistics and machine learning, ensemble methods use multiple learning algorithms to obtain better predictive performance than that which could be obtained from any of the constituent learning algorithms alone.
Model stacking: An ensemble-learning method that involves training a learning algorithm to combine the predictions of several other learning algorithms. First, all other algorithms are trained using available data, and a combiner (master) algorithm is then trained to make a final prediction using all predictions of the other algorithms as additional inputs.
Random search: a family of numerical optimization methods that do not require the gradient of the problem to be optimized, and randomized search can hence be used on functions that are not continuous or differentiable.

## Appendix B. Information about FE Models

Building a good FE model is a long and time-consuming process in which one needs to maintain a balance between calculation time (i.e., attempting to use a small number of elements) and the quality of the calculation results. At the first stage, it is necessary to refine possible flaws in the geometric model and divide it into simple volumes. In the process of numerous refinements, stress-concentration zones are defined, and simple volumes are allocated for building a high-quality mesh. An example of improving the mesh in the air zone at the output edge is shown in Figure A1. Selected volumes for improving the mesh are shown in red.

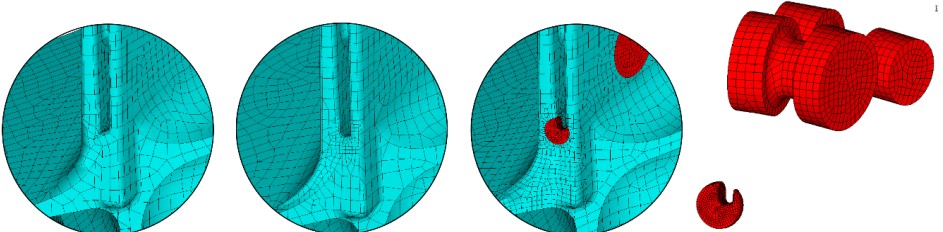

**Figure A1.** Example of mesh refinement in stress-concentration area.

To assess the quality of an FE model, in addition to evaluating the geometric shape of finite elements, which is performed automatically by the ANSYS preprocessor, a special quality criterion was used. The criterion is the ratio of the maximal von Mises stress difference between nodes adjacent to the node under consideration to the average von Mises stress at this node. This criterion is determined for each node on the basis of calculation results (in linear formulation) of the part under consideration with all external loads acting on it. To achieve correct results, the value of this criterion should be less than 5–10%. In zones of contact pairs and in obviously "noncritical" zones, the value of the criterion may exceed this value. Figure A2 shows in the colored areas where the quality criterion exceeded 10%. Analysis of the results presented in Figure A2 showed that the quality of the FE model in the most loaded zones met the requirements.

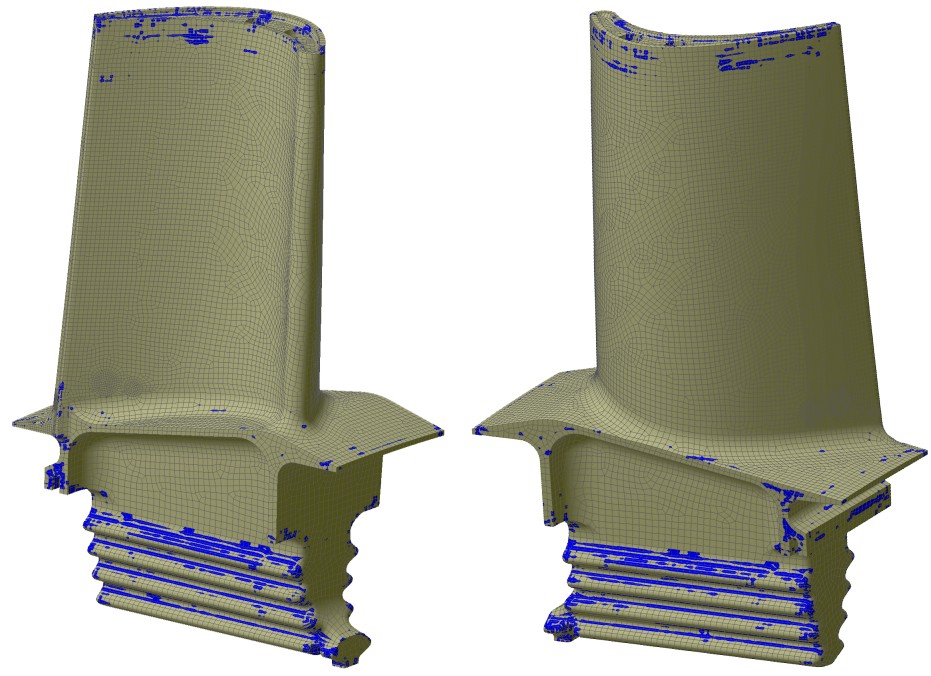

**Figure A2.** Mesh quality.

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
