# Peer review of "Residual Life Prediction of Gas-Engine Turbine Blades Based on Damage Surrogate-Assisted Modeling"

_applsci, doi:10.3390/app10238541_

Round 1

Reviewer 1 Report

The paper presents a study on residual life estimation of gas turbine blades that can improve the decision making process of operators.The topic of the paper is of interest to the gas turbine community.The paper is well written and organized. I recommend the paper for publication.

Author Response

Dear Reviewer, 

Thank you for reviewing our paper. We hope it will be of much value to the gas turbine engineering community.

Reviewer 2 Report

  1. Summary:

The manuscript presents a methodology in which surrogate-based machine learning techniques are used to estimate the remaining life of a turbine blade. The approach was used to simulate the gas turbine engine power plant in two Russian cities. The results of the surrogate model appeared to be properly validated against true data.

   2. Comments:

The topic is interesting and the manuscript reads well. The description of the content is in general clear. However, the results  should have been further supported and integrated with additional information regarding the equations that define the physics of the system under consideration. I suggest to include some of this information as appendices. More detailed information about the finite element modelling should also be added.  It must include the details of how the finite element figures are obtained (e.g., software used, etc). Details about the hyper parameters must be also added.

   3. Recommendation:

As mentioned above, the study is interesting. However, at this stage, the manuscript needs a few improvements in terms of presentation of the content.  Those improvements must be properly addressed.  

Author Response

Dear Reviewer,

Thank you for reviewing our paper

Please see the response in the attachment

Reviewer 3 Report

In my opinion it is a quality work that deserves publication.

Information on the numerical models used in the simulations would be welcome as they would allow a better understanding of the dimension of the problem.

How critical is the influence of the quality of the numerical models on the results of the proposed model?

Author Response

Dear Reviewer

Thank you for reviewing our paper

Please see the response in the attachment

This manuscript is a resubmission of an earlier submission. The following is a list of the peer review reports and author responses from that submission.